# Electrocatalytic Isomerization of Allylic Alcohols: Straightforward Preparation of β-Aryl-Ketones

Anding Li [1,†], Nan Zheng [1,†], Kai Guo [1], Zhongchao Zhang [1] and Zhen Yang [1,2,3,*]

1   State Key Laboratory of Chemical Oncogenomics and Key Laboratory of Chemical Genomics, Peking University Shenzhen Graduate School, Shenzhen 518055, China; lianding@pku.edu.cn (A.L.); zhengnan123@pku.edu.cn (N.Z.); guok@pku.edu.cn (K.G.); zc_zhang@pku.edu.cn (Z.Z.)
2   Key Laboratory of Bioorganic Chemistry and Molecular Engineering of Ministry of Education and Beijing National Laboratory for Molecular Science (BNLMS), and Peking-Tsinghua Center for Life Sciences, Peking University, Beijing 100871, China
3   Shenzhen Bay Laboratory, Shenzhen 518055, China
*   Correspondence: zyang@pku.edu.cn
†   These authors have contributed equally to this work.

**Abstract:** Electrochemical synthesis has been rapidly developing over the past few years. Here, we report a practical and eco-friendly electrocatalytic isomerization of allylic alcohols to their corresponding carbonyl compounds. This reaction can be carried out in undivided cells without the addition of external chemical oxidants and metal catalysts. Moreover, this reaction features a broad substrate scope including challenging allylic alcohols bearing tri- and tetra-substituted olefins and affords straightforward access to diverse β-aryl-ketones. Mechanistic investigations suggest that the reactions proceed through a radical process. This study represents a unique example in which electrochemistry enables hydrogen atom transfer in organic allylic alcohol substrates using a simple organocatalyst.

**Keywords:** electrocatalytic; isomerization; allylic alcohols; Hydrogen Atom Transfer (HAT)



## 1. Introduction

The prevalence and biological importance of carbonyl compounds in the pharmaceutical, fine chemical, and materials science industries have driven a growing interest in developing more efficient methodologies for their synthesis [1]. In this context, isomerization has emerged in recent years as an attractive and straightforward strategy for synthesizing versatile, challenging, substituted carbonyl compounds from readily available or easily accessible allylic alcohols [2–6]. Compared with conventional conversions involving stepwise transformations that often require stoichiometric external oxidants and reductants, the isomerization of allylic alcohols to their corresponding carbonyl compounds has significant advantages in the atom and step economy [7–10]. In recent decades, numerous approaches have been developed for the catalytic isomerization of allylic alcohols based on metal catalysts involving iron [11], rhodium [12,13], ruthenium [14–16], iridium [17,18], palladium and others in one-pot, redox-neutral processes (Scheme 1A) [19–21]. These methods enabled the rapid assembly of versatile carbonyl compounds from readily accessible allylic alcohols via a single-step isomerization process. Although great success has been achieved, these transformations proved to be less efficient when using heavily substituted olefins, particularly those with tetra-substituted double bonds that are sterically hindered for binding and reacting with metal catalysts [22,23]. Recently, the photoredox-mediated hydrogen atom transfer (HAT) process has demonstrated exciting results for C-H bond functionalization, as well as other transformations [24–32]. We previously described a photoredox-catalyzed isomerization of γ-carbonyl-substituted allylic alcohols to their corresponding 1,4-dicarbonyl compounds by combining photoredox and HAT catalysts (Scheme 1B) [33]. Further theoretical studies indicated that electron-deficient

$\gamma$-carbonyl substituents were crucial to the single-electron reduction process. Despite the great advances in generating functionalized 1,4-dicarbonyl compounds, this method failed to catalyze the isomerization of allylic alcohols bearing other substituent groups at the $\gamma$ position in our initial attempts (such as $\gamma$-aryl-allylic alcohols), which largely restricted its application (Scheme S1).

**A) Metal-catalyzed isomerization of allylic alcohols**

**B) Our previous work: Photoredox-catalyzed isomerization of allylic alcohols**

**C) This work: Electrocatalytic isomerization of allylic alcohols**

**Scheme 1.** Metal-catalyzed (**A**), photoredox-catalyzed (**B**), and electrocatalytic isomerization (**C**) of allylic alcohols to their corresponding carbonyl compounds.

In order to expand the application scope and develop a more efficient synthetic methodology, we became interested in electrochemistry [34–38]. Electrochemical transformations have emerged as a useful technology for achieving efficient and sustainable methodologies for organic synthesis, using electrons as direct and traceless redox reagents, thus decreasing the use of expensive metal catalysts and additives [39–50]. We envisioned that combining direct electrolysis with the HAT process would be an effective strategy to realize the isomerization of substituted $\gamma$-aryl-allylic alcohols [51–56]. In this work, we have developed an electrocatalytic isomerization approach to achieve the straightforward synthesis of diverse $\beta$-aryl-ketones, which are widely distributed as core scaffolds in bioactive compounds and natural products (Figure 1).

**Figure 1.** Representative bioactive molecules containing $\beta$-aryl-ketones.

## 2. Results and Discussion

At the outset, we chose $\gamma$-phenyl allylic alcohol **1a**, which was easily prepared from cyclohexanone via a two-step transformation, as a model substrate to optimize the electrocatalytic isomerization conditions (Table 1). A 72% yield of the desired $\beta$-phenyl hexanone **2a** was obtained by using a two-electrode system in an undivided cell (glassy carbon rod

anode and platinum plate cathode, 8 mA constant) at room temperature, with quinuclidine (20 mol%) as the catalyst, $^n$Bu$_4$NH$_2$PO$_4$ (1.0 equiv.) as the electrolyte (Table 1, entry 1). Both electricity (Table 1, entry 2) and the organic catalyst (Table 1, entry 3) were critical for the formation of **2a**. Conducting the reaction at a current of 5 mA (Table 1, entry 4) led to a reduced yield of 58%, while a higher constant current of 12 mA (Table 1, entry 5) resulted in substrate decomposition. Although applied extensively as an HAT mediator, DABCO was found to be ineffective in promoting the formation of **2a** in the current system (Table 1, entry 6) [54,57]. The yield of **2a** was also slightly diminished to 62% by changing the solvent to pure MeCN (Table 1, entry 7). However, other solvents, including DMF (Table 1, entry 8) and HIFP (Table 1, entry 9), led to only a trace amount of product. When the reaction was carried out in the presence of water, no desired product could be detected (Table 1, entry 10). Changing the supporting electrolyte to $^n$Bu$_4$NBF$_4$ (Table 1, entry 11) and $^n$Bu$_4$NBF$_6$ (Table 1, entry 12) resulted in reduced reaction efficiency. Importantly, no product could be obtained under an air atmosphere (Table 1, entry 13), likely due to the oxidative decomposition of quinuclidine. Finally, lower yields were observed with an alternative zinc plate cathode (Table 1, entry 14) and glassy carbon rod cathode, indicating the importance of platinum cathode (Table 1, entry 15). Alight drop in yield was observed with an extended reaction time (Table 1, entry 16).

**Table 1.** Optimization of the reaction conditions [1].

| Entry | Variation from the Standard Conditions | Yield (%) [2] |
|---|---|---|
| 1 | none | 72 |
| 2 | no electricity | n.d. (100) |
| 3 | no quinuclidine | n.d. (90) |
| 4 | 5 mA | 15 (59) |
| 5 | 12 mA | n.d. |
| 6 | DABCO instead of quinuclidine | n.d. (95) |
| 7 | MeCN as solvent | 62 |
| 8 | DMF as solvent | trace (80) |
| 9 | HIFP as solvent | trace (42) |
| 10 | solvent containing 0.5 mL H$_2$O | n.d. (98) |
| 11 | $^n$Bu$_4$NBF$_4$ instead of $^n$Bu$_4$NH$_2$PO$_4$ | 31 (50) |
| 12 | $^n$Bu$_4$NPF$_6$ instead of $^n$Bu$_4$NH$_2$PO$_4$ | n.d. (95) |
| 13 | under Air | trace (55) |
| 14 | Zn(-) instead of Pt(-) | 29 |
| 15 | Glassy carbon(-) instead of Pt(-) | trace |
| 16 | reaction time to 24 h | 70 |

[1] Standard reaction conditions: undivided cell, glassy carbon rod anode (2 × 8 × 52 mm$^3$), platinum plate cathode (2 × 8 × 52 mm$^3$), constant current = 8 mA, **1a** = 0.5 mmol, $^n$Bu$_4$NH$_2$PO$_4$ = 0.5 mmol, quinuclidine = 20 mol%, MeCN/toluene (5:1) = 6 mL, under argon, room temperature, 12 h. [2] Isolated yield. Recovery of unreacted **1a** shown in parentheses. n.d. = not detected. DABCO = 1,4-diazabicyclo [2.2.2]octane. MeCN = acetonitrile. DMF = dimethylformamide. HIFP = hexafluoroisopropanol.

With the optimized conditions in hand, we then moved on to evaluating the substrate scope. To this end, various substituted γ-aryl-allylic alcohols were rapidly synthesized from readily available starting materials in 3–4 step transformations and then tested to extend the application scope (Figure 2). To our delight, a number of tri-substituted secondary allylic alcohols bearing different carbon skeletons as well as electron-rich and electron-deficient substituents on the phenyl unit were well-tolerated in this process, and could afford the corresponding β-aryl-ketones in moderate to good yields (**2a–2j**), except for **2h** which may be due to the oxidative sensitivity within the pyridine moiety. The significant drop in

the yield of **2k** was probably due to steric hindrance, which reduced the accessibility of the quinuclidinium radical cation to the olefins. Remarkably, using the standard reaction conditions, the isomerization of allylic alcohols with tetra-substituted olefins successfully provided the corresponding α,β-bis-substituted ketones in moderate to good yields (**2l–2s**). The obtained trans/cis ratios in these compounds may contribute to the protonation and tautomerization of intermediates in the reaction process. As a result, our currently developed methodology might provide a practical means of synthesizing these synthetically challenging motifs.

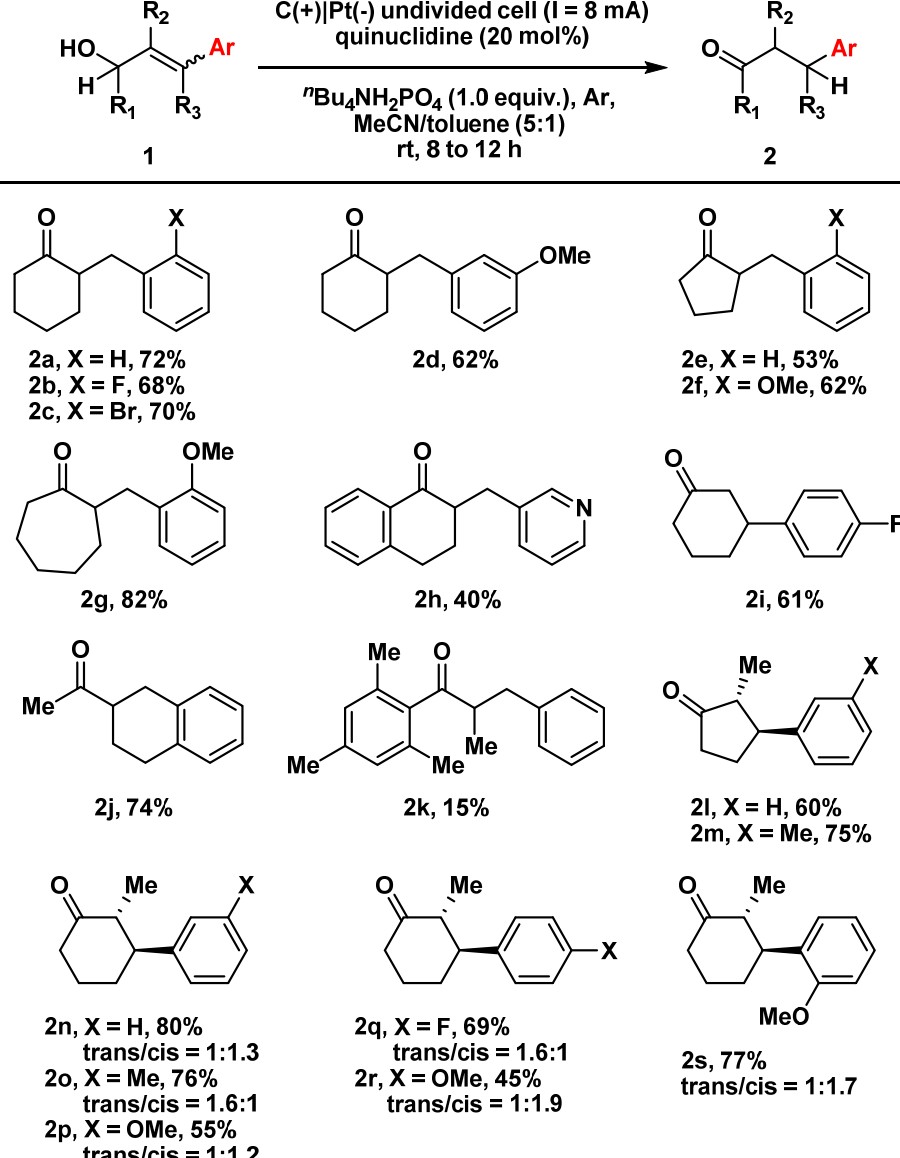

**Figure 2.** Substrate scope of γ-aryl-allylic alcohols. Reactions were conducted at a 0.5 mmol scale using standard reaction conditions. All tri-substituted allylic alcohol substrates were E-isomers. The trans/cis ratios of products were determined by [1]H-NMR spectra.

To verify the practical feasibility of our electrocatalytic isomerization process, a 10-fold scaling-up synthesis of **2b** (5.0 mmol) was conducted using a modified electrolysis device and reaction condition (Scheme 2, Supporting Information for reaction device setup). The desired isomerization proceeded smoothly, allowing for the large-scale synthesis of **2b** in 78% yield (92% brsm). This result further demonstrated the potential application of this isomerization protocol in pharmaceutical and synthetic research.

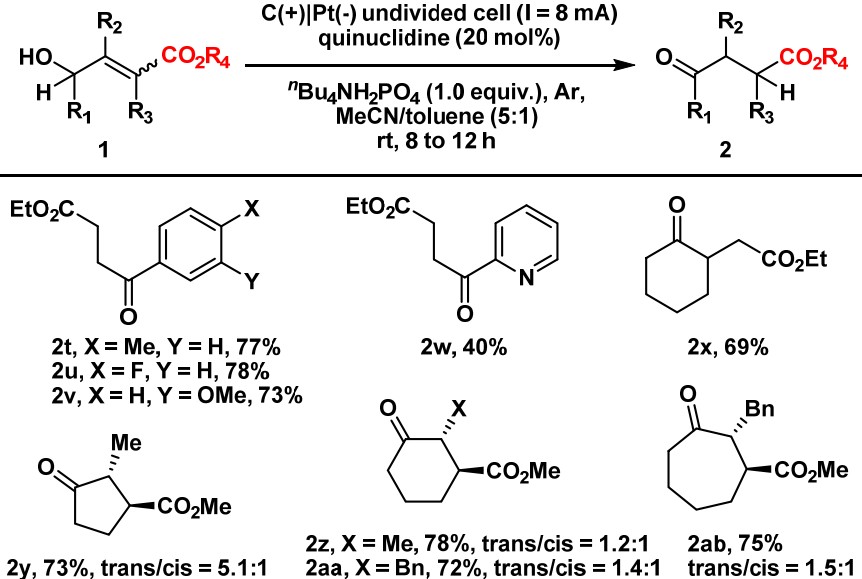

**Scheme 2.** Large-scale experiment. Reaction conditions: Undivided cell, Graphite rod anode (30 × 30 × 3 mm³), Platinum plate cathode (30 × 30 × 0.1 mm³), constant current = 20 mA, **2a** = 5.0 mmol, $^nBu_4NH_2PO_4$ = 5.0 mmol, quinuclidine = 30 mol%, MeCN/toluene (5:1) = 50 mL, under argon, room temperature, 24 h. brsm = based on recovered starting material.

Furthermore, substrate scope investigation using γ-carbonyl allylic alcohols as substrates also demonstrated its broad reaction compatibility, even with tetra-substituted olefins, affording the corresponding β-carbonyl ketones that are important intermediates in organic synthesis (Figure 3). Notably, compared with our previously developed photoredox-catalyzed isomerization method that required a longer reaction time (up to 5 days at 40 °C) [33], the electrocatalytic isomerization strategy exhibited significantly enhanced reaction efficiency and higher energy utilization (within 12 h at room temperature).

**Figure 3.** Substrate scope of γ-carbonyl-allylic alcohols. Reactions were conducted at a 0.5 mmol scale using standard reaction conditions. All tri-substituted allylic alcohol substrates were E-isomers. The trans/cis ratios of the products were determined by ¹H-NMR spectra.

To gain insights into the reaction mechanism, control experiments were performed (Figure 4A). Treatment with additive TEMPO (2.0 equiv.) dramatically diminished the reaction efficiency, which supported a radical process as the main pathway. This hypothesis was further supported by the successful detection of the TEMPO-adducted product **TEMPO-2a** via HRMS. Subsequently, the isomerization of the deuterium labeled substrate *d*-**1a** under the optimized conditions resulted in unlabeled product **2a**, suggesting an intermolecular protonation process that is consistent with our previous discovery [33]. The decreased yield was likely due to the isotopic effect, which caused a reduced efficiency of the HAT process [58,59]. The catalytic role of quinuclidine in oxidizing the allylic alcohol substrate **1a** was further confirmed by a cyclic voltammetry (CV) study (Figure 4B). The oxidation of quinuclidine would be preferred at a lower potential (E = 1.59 V vs. SCE), which is below that of substrate **1a** (E = 1.68 V vs. SCE).

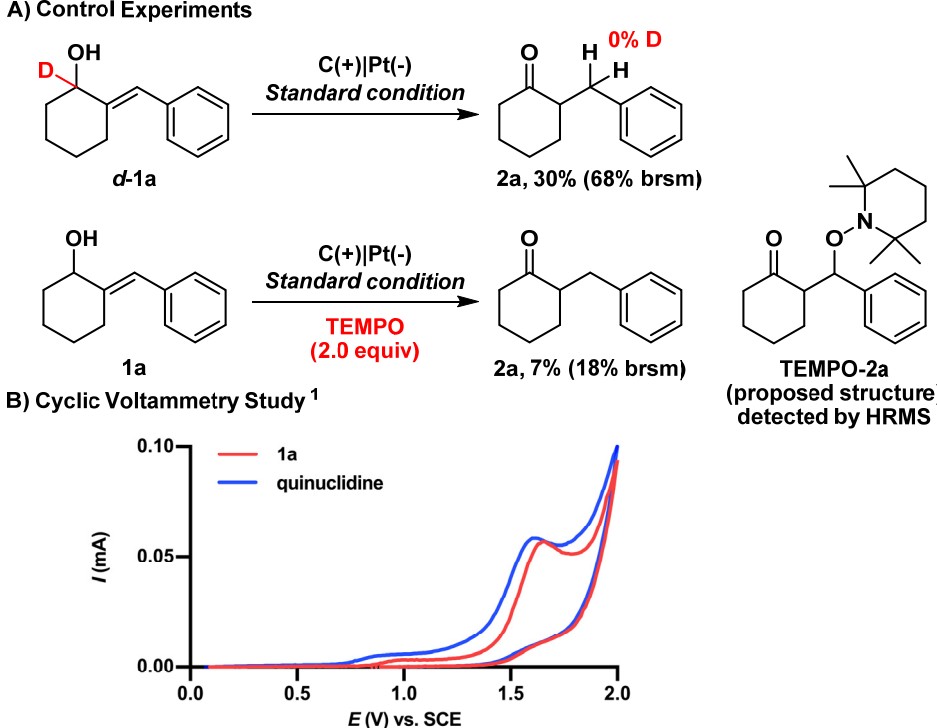

**Figure 4.** Mechanism studies. (**A**) Deuterium label experiment and radical trapping experiment. (**B**) Cyclic voltammetry study. [1] Cyclic voltammograms of substrate **1a** and quinuclidine in dry and deoxygenated MeCN/toluene (5:1) containing $^{n}Bu_4NH_2PO_4$ (1.0 equiv.) and a potential sweep rate of 100 mV/s (vs. SCE). brsm = based on recovered starting material.

Finally, based on the above results and our previous study, a plausible mechanism was proposed (Figure 5). The electrolysis begins with the anodic oxidation of quinuclidine [q] to generate its radical cation $[q]^+$, which then initiates the hydrogen atom transfer (HAT) process with the allylic alcohol hydrogen-bonded species **ii** (phosphonate product of starting material **i**) to form the corresponding allylic alcoholic radical **iii,** along with the quinuclidinium cation $[q]^{H+}$. The alcoholic radical **iii** is further reduced to the phosphonate allylic alcohol anion **iv** at the cathode and quenched to enol **v** via protonation, which eventually affords ketone **vi** via tautomerization.

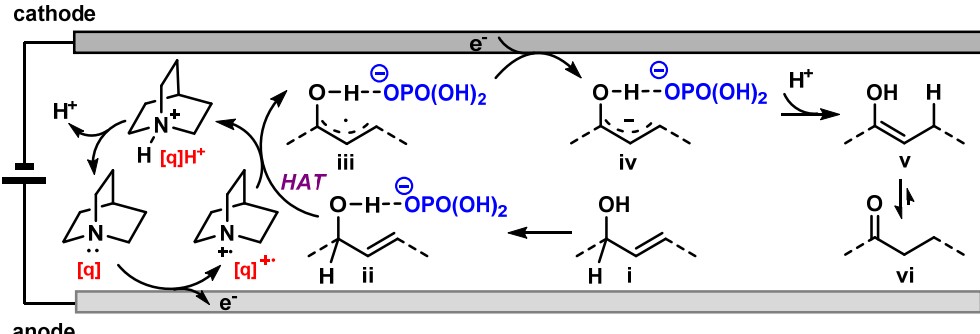

**Figure 5.** Proposed mechanism for the electrocatalytic isomerization of allylic alcohols.

## 3. Materials and Methods

### 3.1. General Information

Unless otherwise stated, all chemicals were purchased at the highest commercial quality (>95%) and used directly without purification. Anhydrous tetrahydrofuran (THF) and toluene were distilled from sodium-benzophenone, and MeCN was distilled from

calcium hydride. All reactions were carried out in flame-dried flasks with magnetic stirring under an argon gas atmosphere with dry solvents (exceptions were made for condition optimization entries where air and water were added intentionally). External oil-baths were used to record all reaction temperatures. Low-temperature reactions were conducted in a Dewar vessel filled with acetone/dry ice ($-78$ °C) or distilled water/ice (0 °C). Reactions were monitored by thin-layer chromatography (TLC) carried out on 0.25 mm Tsingdao silica gel glass-backed plates (60F-254) and visualized under UV light at 254 nm. Staining was performed with an ethanolic solution of phosphomolybdic acid (PMA) and heat as developing agents. Tsingdao silica gel (60, particle size 0.040–0.063 mm) was used for flash column chromatography for product purification. Yields refer to chromatographical isolation unless otherwise specified. NMR spectra were recorded on Brüker Advance 500, 400 and 300 MHz and reported as follows: chemical shift $\delta$ in ppm (multiplicity, coupling constant $J$ in Hz, number of protons) for $^1$H NMR spectra and chemical shift $\delta$ in ppm for $^{13}$C NMR spectra. The following abbreviations were used to explain the multiplicities: s = singlet, d = doublet, t = triplet, q = quartet, m = multiplet, or combinations thereof. Residual solvent peaks in $CDCl_3$ ($\delta$ H = 7.26 ppm, $\delta$ C = 77.16 ppm), $CD_2Cl_2$ ($\delta$ H = 5.32 ppm, $\delta$ C = 53.84 ppm), $CD_3OD$ ($\delta$ H = 3.31 ppm, $\delta$ C = 49.00 ppm) was used as an internal standard reference. High-resolution mass spectrometric (HRMS) data were recorded on Q-TOF-MS instruments (Brüker Apex IV RTMS and VG Auto Spec-3000). Melting points (m.p.) were recorded on a SGWX-4B apparatus. Cyclic voltammetry (CV) analysis was recorded at CHI 660D Instrument.

### 3.2. General Experimental Procedure for Electrocatalytic Isomerization of Allylic Alcohols

Quinuclidine (20 mol%, 12 mg), and $^n$Bu$_4$NH$_2$PO$_4$ (0.5 mmol, 170 mg). MeCN (5 mL), toluene (1 mL) were added to the flame-dried and argon charged ElectraSyn 2.0 reaction vial (IKA, 10 mL) with a stir bar. Next, MeCN (5 mL), toluene (1 mL) was added, followed by the corresponding allylic alcohol substrate (0.5 mmol, 1.0 equiv.). The EletraSyn 2.0 vial cap equipped with the anode (glassy carbon, $2 \times 8 \times 52$ mm$^3$) and cathode (platinum, $2 \times 8 \times 52$ mm$^3$) were inserted into the mixture. The reaction system was strictly deoxygenated several times with argon. The reaction mixture was then electrolyzed under a constant current of 8 mA for 8–12 h under argon. When the reaction was completed as indicated from the TLC check, the mixture was collected and electrodes were rinsed several times with ethyl acetate. The mixture was filtered through a plug of celite and washed with ethyl acetate. The filtrate was concentrated under reduced pressure, and the residue was purified by flash column chromatography on silica gel to provide the product.

### 4. Conclusions

In summary, we developed an electrocatalytic isomerization of allylic alcohols, which provides straightforward and eco-friendly access to functionalized β-aryl and β-carbonyl ketones, with good yields. From a synthetic point of view, this protocol displays a broad substrate scope and obviates the requirement of chemical oxidant reagents and expensive metal catalysts. Additionally, a plausible hydrogen atom transfer (HAT) mechanism was proposed based on control experiments and the cyclic voltammetry (CV) study. Finally, the successful use of this protocol in the scale-up experiment further highlights its potential application in organic synthesis. Further synthetic applications, as well as investigations into the enantioselective version of this reaction, are currently in progress in our laboratory.

**Supplementary Materials:** The following Supporting Information can be downloaded at: https://www.mdpi.com/article/10.3390/catal12030333/s1, Scheme S1: Initial attempts of photoredox-catalyzed isomerization of **1a**; Reaction setup for electrolysis; Preparation of the substrates; Mechanistic studies; Characterization data of compounds; NMR spectra of compounds.

**Author Contributions:** Conceptualization, A.L., N.Z., K.G., Z.Z., and Z.Y.; methodology, A.L., N.Z., K.G. and Z.Z.; investigation, A.L., N.Z. and K.G.; resources, Z.Y.; data curation, A.L. and N.Z.; writing—original draft preparation, N.Z., A.L., Z.Y.; writing—review and editing, N.Z., A.L., Z.Y.;

visualization, N.Z., Z.Y.; supervision, Z.Y.; funding acquisition, Z.Y., A.L. and N.Z. have contributed equally to this work. All authors have read and agreed to the published version of the manuscript.

**Funding:** This research was financially funded by National Science Foundation of China (Grant Nos. 21632002, 21772008, 21871012, 22171013), Guangdong Natural Science Foundation (Grant No. 2020B0303070002), Shenzhen Basic Research Program (Grant No. JCYJ20170818090044432).

**Data Availability Statement:** Not applicable.

**Acknowledgments:** The authors thank Dian Li and Xin-peng Mu for helpful discussion.

**Conflicts of Interest:** The authors declare no conflict of interest.

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
