# Peer review of "Electrocatalytic Isomerization of Allylic Alcohols: Straightforward Preparation of β-Aryl-Ketones"

_catalysts, doi:10.3390/catal12030333_

Round 1

Reviewer 1 Report

The manuscript by Yang and co-workers describes a very interesting report on allylic alcohol isomerization induced by the electrolytic system. This is a promising, nice extension of their project (see a previous paper, Angew. Chem. Int. Ed. 2020, 59, 11660.) to realize the metal catalyst free isomerization. The target ketones were obtained in good yields from various allylic alcohols with preservation of the functional groups. Other merits of this reaction system are the expanded utility of the substrates such as cyclic tetra-substituted allylic alcohol, short reaction time, enhanced reaction efficiency, and higher energy utilization. The synthetic scope and advantages are, what I believe, in accord with required standard of synthetic work. I recommend provisional acceptance of this work in Catalysts journal, after addressing the following issues;

1) In Figure 2 and 3, E/Z structure of the starting allylic alcohols 1 are not clear in some cases, i.e., 1a-h, 1k, and 1t-x. Also, the general structure of alcohol 1 and ketone 2 suggested in the reaction scheme do not reflect all the substrate and product structures.  

2) The observed trans/cis ratio of the cyclic products in Figure 2 and 3 should be explained.

3) In Figure 4, TEMPO would inhibit the reaction, but its precise inhibition mechanism is not clear. If applicable, the authors should try to isolate TEMPO-derived product, and discuss the inhibition mechanism.                 

4 What is the meaning of ‘‘electrocatalytic’’ in this study? It is not clear the reactions completed with ‘‘catalytic amount of electricity’’.

5) In Scheme 2, the numbering of the starting material is wrong.

Author Response

Reviewer 1:

Q1: In Figure 2 and 3, E/Z structure of the starting allylic alcohols 1 are not clear in some cases, i.e., 1a-h, 1k, and 1t-x. Also, the general structure of alcohol 1 and ketone 2 suggested in the reaction scheme do not reflect all the substrate and product structures.

R1: We apologize for being unclear. All tri-substituted allylic alcohol substrates (1a-h, k) were E-isomers, which was controlled by the first step aldol reaction in substrate synthesis that provided the E-olefin enone as major product (>15:1) (Supporting information). We have added this information in the Figure 2 and Figure 3 legends as “All tri-substituted allylic alcohol substrates were E-isomers.” We have also updated the general structure of 1 and 2 to cover all substrate types.

Q2: The observed trans/cis ratio of the cyclic products in Figure 2 and 3 should be explained.

R2: We thank the reviewer for this question. From the proposed mechanism, the trans/cis isomer was potentially controlled by two parts. 1) Protonation at beta position 2) Tautomerization of enolate. We have added comments regarding this point in the text (line 110-111). “The obtained trans/cis ratios in these compounds may attribute to protonation and tautomerization of intermediates in the reaction process.

Q3: In Figure 4, TEMPO would inhibit the reaction, but its precise inhibition mechanism is not clear. If applicable, the authors should try to isolate TEMPO-derived product, and discuss the inhibition mechanism.                 

R3: We thank the reviewer for this discussion. Actually, we have tried but unfortunately failed to isolate any TEMPO-derived products. We believe it may be extremely unstable for isolation. However, we were able to detect this product via HRMS ([M+H]+ calc 344.2584; found ) this result was shown in the Supporting Information. Based on this we updated the Figure 4A and text (line 147-149) as “This hypothesis was further supported by a successful detection of the TEM-PO-adducted product TEMPO-2a via HRMS.

Q4: What is the meaning of ‘‘electrocatalytic’’ in this study? It is not clear the reactions completed with ‘‘catalytic amount of electricity’’.

R4: We apologize for being unclear. In this study, the phrase ‘‘electrocatalytic’’ refers to the electrochemical organic transformation by using catalytic amount of organocatalyst (20 mol% quinuclidine). The redox organocatalyst functions as mediate which can be used to shuttle electrons and holes from the electrodes to substrates in solution when the direct electrolysis kinetics are inefficient. This concept can be further supported by following papers: J. Org. Chem. 2021, 86, 16001-16007; Org. Lett. 2018, 20, 361-364; ACS Sustainable Chem. Eng. 2021, 9, 1932-1940.

Q5: In Scheme 2, the numbering of the starting material is wrong.

R5: We have corrected this mistake by changing 2a to 1b.

Reviewer 2 Report

The authors present their manuscript describing the synthesis of beta-aryl-ketones via the electrocatalytic isomerization of allylic alcohols.  The manuscript is well-written and the developed method will be of interest to synthetic chemists.  I have a few suggestions for the authors to consider that I believe will improve the manuscript:

  1. It would be helpful to comment on the ready availability of the starting allylic alcohols either in the introduction or at the beginning of the results and discussion.
  2. Scheme 2: "brsm" should be defined as it is not a common abbreviation (I assume it is "based on recovered starting material").
  3. Figure 5:  I wonder about the feasibility of structure iv.  I suspect that even if formed momentarily that the proton on the OH group would be transferred to the negatively charged carbon (to form the enolate).  Perhaps a better initially-formed intermediate is the enolate itself without proceeding through iv.
  4. For the products in which cis/trans isomers were formed from the reaction, something should be said about how the isomers were differentiated and assigned (based on literature spectra? based on NMR spectra?).

Once the above have been considered by the authors and addressed in a revised version I would support publication of this paper.

Author Response

Reviewer 2:

Q1; It would be helpful to comment on the ready availability of the starting allylic alcohols either in the introduction or at the beginning of the results and discussion.

R1: We thank the reviewer for the advice. All starting allylic alcohols can be prepared easily from commercially available chemicals. We have revised the text (line 64-65) as follows “At the outset, we chose γ-phenyl allylic alcohol 1a, which was easily prepared from cyclohexanone via a two-step transformation, as a model substrate to optimize the electrocatalytic isomerization conditions (Table 1).” Also, (line 92-95) as follows to comment on this point. “With the optimized conditions in hand, we then moved on to evaluate the substrate scope. To this end, various substituted γ-aryl-allylic alcohols were rapidly synthesized from readily available starting materials in 3-4 steps’ transformations and then tested to extend the application scope (Figure 2).”

Q2: Scheme 2: "brsm" should be defined as it is not a common abbreviation (I assume it is "based on recovered starting material").

R2: We thank the reviewer for this advice. We have added this information in the legend of Scheme 2.

Q3: Figure 5: I wonder about the feasibility of structure iv. I suspect that even if formed momentarily that the proton on the OH group would be transferred to the negatively charged carbon (to form the enolate). Perhaps a better initially-formed intermediate is the enolate itself without proceeding through iv.

R3: We thank the reviewer for the discussion. We agree that intermediate iv is fragile. However, we argue that in this case nBu4NH2PO4 (1.0 equiv.) may function as both electrolyte but also a H-bonding forming reagent to drive the reaction. As concluded from Table 1, the use of nBu4NH2PO4 is crucial to the reaction whereas nBu4NBF4 and nBu4NPF6 significantly lower down the yield. Therefore, we revised intermediate iv in Figure 5 to the -NHPO4 bonded form which can be further quenched by external proton.

Q4: For the products in which cis/trans isomers were formed from the reaction, something should be said about how the isomers were differentiated and assigned (based on literature spectra? based on NMR spectra?).

R3: We apologize for being unclear. The trans/cis ratio of the α,β-bis-substituted ketone products were determined by 1H-NMR spectra. The chemical shifts of the β-H in these molecules are typical for isomer identification. Besides, some of our products were known compounds from previous reports which were assigned before (J. Org. Chem. 2003, 68, 9226–9232; Synlett, 2017, 28, 2829-2832). We have updated this information in the Figure 2 and Figure 3 legends as “The trans/cis ratio of products were determined by 1H-NMR spectra.”.

Reviewer 3 Report

The authors of the publication "Electrocatalytic Isomerization of Allylic Alcohols: Straightforward Preparation of β-Aryl-Ketones" presented a study on electrocatalytic isomerization of allyl alcohols, which enables the simple and environmentally friendly preparation of β-aryl and β-carbonyl ketone.

The work that was assessed was correctly presented. Meets the requirements for the authors of the Catalysts journal. The authors correctly presented the results of their research.

1) However, the presented work should be supplemented with a more extensive review of the literature. In particular, it begs to develop issues related to the use of catalysts based on various metals: "(...) In recent decades, numerous approaches have been developed for the catalytic isomerization of allylic alcohols based on metal catalysts involving iron, rhodium, ruthenium, iridium, palladium, and others in one-pot redox-neutral processes (Scheme 1A) [11-21]. (...) ". The metals for the catalysts have been mentioned, but what the results of the research showed, nothing on this topic is found in the presented article.

2) Authors should also pay attention to the number of cited references for one sentence. This remark is noticeable in several cases, for example in the sentence quoted above: 10 references were cited to one sentence. Another example is the sentence: "(…) Recently, the photoredox-mediated hydrogen atom transfer (HAT) process has demonstrated exciting results for C-H bond functionalization as well as other transformations [24-32]. (...) ", where 8 references were cited. Yet another example of this remark, which should be cited: "(...) The electrochemical transformations have emerged as a useful technology for achieving efficient and sustainable methodologies for organic synthesis using electrons as direct and traceless redox reagents, thus decreasing the use of expensive metal catalysts and additives [34-50]. (...) ", where as many as 18 references were cited. I believe that categorically this way of citing should be changed.

3) Another remark is related to the lack of citation in the text of Scheme 1c. Perhaps Scheme S1) was incorrectly inserted, which of course should be changed. Similarly, in "Scheme 1. Metal-catalyzed (A), photoredox-catalyzed (B), and electrocatalytic isomerization (C, this work) of allylic alcohols to their corresponding carbonyl compounds." There is no parenthesis after the letter C: there is (C, and it should be (C).

4) The abstract and conclusions should also be improved. In the case of the first part, there is no more detailed information on the content of the entire article. There is only general information that does not necessarily clearly define or emphasize the value of the research described. The situation is similar in INTRODUCTION. The paragraph ending this part of the article should start more or less like this. In this work, we investigated… .. I think it would be a "smooth" and logical transition from INTRODUCTION to EXPERIMENTAL. It would be much easier for the reader to understand what the next part of the article is about, the more that there is too little information on this topic in the abstract.

5) No elaboration of the MeCN abbreviation can be found in the text.

6) Another comment concerns the explanation from the authors whether parameters such as temperature, amount of catalyst and reaction time have an impact on the presented process? If so, what and have any research been conducted in this direction? Can the obtained results be presented?

7) CONCLUSIONS - in the last part of the article, which should be a summary part, which should include the advantages of the proposed method, the most important results of the proposed process are described - this publication lacks this. The authors briefly recalled what the topic was about, which in my opinion is insufficient for the recipient.

Author Response

Review 3:

Q1: However, the presented work should be supplemented with a more extensive review of the literature. In particular, it begs to develop issues related to the use of catalysts based on various metals: "(...) In recent decades, numerous approaches have been developed for the catalytic isomerization of allylic alcohols based on metal catalysts involving iron, rhodium, ruthenium, iridium, palladium, and others in one-pot redox-neutral processes (Scheme 1A) [11-21]. (...) ". The metals for the catalysts have been mentioned, but what the results of the research showed, nothing on this topic is found in the presented article.

R1: We thank the reviewer for this question. We have added more comments on the metal-catalyzed isomerization: both pros and cons. Shown below:

In recent decades, numerous approaches have been developed for the catalytic isomerization of allylic alcohols based on metal catalysts involving iron [11], rhodium [12,13], ruthenium [14-16], iridium [17,18], palladium [23], and others in one-pot re-dox-neutral processes (Scheme 1A) [19-21]. These methods enabled rapid assembly of versatile carbonyl compounds from readily accessible allylic alcohols via a single-step isomerization process. Although great success has been achieved, these trans-formations proved to be less efficient when using heavily substituted olefins, particularly those with tetra-substituted double bonds that are sterically hindered for binding and reacting with metal catalysts [22,23].

Q2: Authors should also pay attention to the number of cited references for one sentence. This remark is noticeable in several cases, for example in the sentence quoted above: 10 references were cited to one sentence. Another example is the sentence: "(…) Recently, the photoredox-mediated hydrogen atom transfer (HAT) process has demonstrated exciting results for C-H bond functionalization as well as other transformations [24-32]. (...) ", where 8 references were cited. Yet another example of this remark, which should be cited: "(...) The electrochemical transformations have emerged as a useful technology for achieving efficient and sustainable methodologies for organic synthesis using electrons as direct and traceless redox reagents, thus decreasing the use of expensive metal catalysts and additives [34-50]. (...) ", where as many as 18 references were cited. I believe that categorically this way of citing should be changed.

R2: We thank the reviewer for this suggestion. However most of these citations belong to previously achieved reports in the general field of electrochemical organic synthesis and should be cited to cover the development of this field. We have re-categorized some of the refs as described below:

We have re-distributed citations to certain metal catalyst referred to in the text (line 32-36). “In recent decades, numerous approaches have been developed for the catalytic isomerization of allylic alcohols based on metal catalysts involving iron [11], rhodium [12,13], ruthenium [14-16], iridium [17,18], palladium [23], and others in one-pot re-dox-neutral processes (Scheme 1A) [19-21]

Text (line 52-56): “In order to expand the application scope and develop a more efficient synthetic methodology, we became interested in electrochemistry [34-38]. The electrochemical transformations have emerged as a useful technology for achieving efficient and sustainable methodologies for organic synthesis using electrons as direct and traceless redox reagents, thus decreasing the use of expensive metal catalysts and additives [39-50].”

Q3: Another remark is related to the lack of citation in the text of Scheme 1c. Perhaps Scheme S1) was incorrectly inserted, which of course should be changed. Similarly, in "Scheme 1. Metal-catalyzed (A), photoredox-catalyzed (B), and electrocatalytic isomerization (C, this work) of allylic alcohols to their corresponding carbonyl compounds." There is no parenthesis after the letter C: there is (C, and it should be (C).

R3: We apologize for being unclear here. Scheme 1C is the representation of this new work; there is no ref for this new achievement. Scheme S1 is a supporting scheme located in the supporting information which displays our initial trials of photoredox-catalyzed isomerization on γ-aryl-allylic alcohols (Please refer to Supporting Information for detail). We have updated the Scheme legend as (C) as suggested.

Q4: The abstract and conclusions should also be improved. In the case of the first part, there is no more detailed information on the content of the entire article. There is only general information that does not necessarily clearly define or emphasize the value of the research described. The situation is similar in INTRODUCTION. The paragraph ending this part of the article should start more or less like this. In this work, we investigated… .. I think it would be a "smooth" and logical transition from INTRODUCTION to EXPERIMENTAL. It would be much easier for the reader to understand what the next part of the article is about, the more that there is too little information on this topic in the abstract.

R4: We thank the reviewer for this advice. We have updated the abstract and text (line 58)

Q5: No elaboration of the MeCN abbreviation can be found in the text.

R5: We have updated the Table 1 legend with solvent abbreviation information.

Q6: Another comment concerns the explanation from the authors whether parameters such as temperature, amount of catalyst and reaction time have an impact on the presented process? If so, what and have any research been conducted in this direction? Can the obtained results be presented?

R6: We thank the reviewer for this question. We did conduct the reaction under 50 oC heating in the large-scale reaction (see supporting information for device) and it turned out heating had no significant influence on the reaction. 20 mol% quinuclidine was the minimal catalyst loading and higher loading (40%) makes no difference in yields. We have updated Table 1 with entry 16 where extended reaction time was performed.

Q7: CONCLUSIONS - in the last part of the article, which should be a summary part, which should include the advantages of the proposed method, the most important results of the proposed process are described - this publication lacks this. The authors briefly recalled what the topic was about, which in my opinion is insufficient for the recipient.

R7: We thank the reviewer for this suggestion. We have added more comments on this section.